

# Improving English-Chinese translation using the Kolmogorov-Arnold Transformer

Yuzhe Nie

School of Foreign Languages, Shanghai University, Shanghai, China

## ABSTRACT

Machine translation is an important part of natural language processing, helping people communicate across languages, localise content, and search for information in different languages. In this article, we introduce a new framework using the Kolmogorov-Arnold Transformer (KAT) to improve translation quality. We test KAT on the Bilingual MELD dataset and compare it with both traditional statistical models and modern neural translation models. Our results show that KAT performs better, achieving a BLEU-4 score of 42.8, a METEOR score of 45.3, and a TER of 40.5, all of which are improvements over standard transformer models. We also find that using a larger vocabulary and adding the Kolmogorov-Arnold network helps improve translation accuracy. These results suggest that Kolmogorov-Arnold-based methods can be a valuable addition to machine translation systems.

## INTRODUCTION

In today's globalised world, effective communication across languages is essential in many domains, including international business, education, and technology. English–Chinese translation plays a crucial role in enabling not only trade but also cross-cultural academic exchange. As China's global influence grows, the demand for high-quality English–Chinese translation has risen, making it a key skill for fostering cooperation and mutual understanding (*Xue & Wang, 2023*; *Fan, 2024*). In the business sector, accurate translation is vital for contract negotiations, product marketing, and partnership development, where miscommunication can result in financial losses and misunderstandings (*Chen, 2020*; *Lin & Chen, 2023*). In education, translation helps disseminate research findings and supports academic collaboration between Chinese- and English-speaking institutions (*Xue & Wang, 2023*; *Fan, 2024*).

However, translating between English and Chinese is a complex task. English is a subject-prominent language with a linear structure, while Chinese is topic-prominent and often presents ideas in a more circular or spiral form (*Jun, 2019*). This fundamental difference can lead to errors such as literal word-for-word translation, which often distorts the original meaning and produces unnatural phrasing in the target language (*Deng & Xue, 2017*). Additionally, the cultural context embedded in language can be lost in translation,

Corresponding author
Yuzhe Nie, nieyuzhe1900@163.com

requiring translators to possess not only strong linguistic skills but also cultural awareness (*Tian, Wong & Bin Abdullah, 2024*; *Wang & Gu, 2016*). These challenges highlight the importance of expert bilingual data to support high-quality translation.

Despite notable advances in translation technologies, challenges remain in English–Chinese translation. Literal translations that ignore grammar and meaning frequently result in incoherent or awkward outputs (*Deng & Xue, 2017*). While machine translation (MT) tools offer convenience, they often fail to capture subtle nuances, leading to technically correct but contextually inappropriate translations (*Gao, 2024*; *Jiang, 2023*). Overreliance on such tools can perpetuate errors and misunderstandings, underscoring the continuing need for skilled human translators.

Recent developments in machine learning and deep learning have significantly improved translation systems. These technologies allow models to learn from vast amounts of bilingual data, gradually enhancing their performance (*Bi, 2020*; *Guo, 2021*). Additionally, the growth of digital platforms such as social media provides rich sources of linguistic data that can be used to extract bilingual text pairs for analysing common expressions and contextual usage (*Sun, 2023*). Studies show that using domain-specific bilingual data can greatly improve translation accuracy and fluency (*Chen & Yu, 2013*; *Bi, 2020*).

Machine Translation has evolved through several stages: Rule-Based Machine Translation (RBMT), Statistical Machine Translation (SMT), and Neural Machine Translation (NMT). Each stage has contributed to the development of systems capable of handling structurally different language pairs like English and Chinese. RBMT, which relies on manually crafted grammar rules and dictionaries, was too rigid and struggled with the complexity of natural language (*Wang, 2024*). SMT marked a turning point by using bilingual corpora to automatically learn translation patterns based on statistical co-occurrence, improving fluency and accuracy (*Kumar et al., 2010*). NMT further advanced the field by using deep neural networks to model translation as an end-to-end process, capturing long-range dependencies and better handling differences in word order and syntax between languages (*Miao et al., 2021*).

Although NMT has led to significant improvements, it still faces difficulties with complex grammatical structures, particularly when translating between languages with different syntax, such as English and Chinese. As a result, hybrid approaches that combine the statistical strengths of SMT with the contextual depth of NMT are gaining attention as promising strategies for achieving more natural and accurate translations (*Wang & Gu, 2016*).

To address the persistent challenges in English–Chinese translation, this study proposes an efficient hybrid model called the Kolmogorov-Arnold Transformer (KAT). By integrating the structured decomposition capabilities of Kolmogorov-Arnold Networks into the Transformer architecture, KAT enhances both the expressiveness and efficiency of neural machine translation systems. This approach not only improves parameter efficiency and translation fluency but also demonstrates superior performance across multiple standard metrics. The proposed model aims to advance the state of bilingual translation

systems, particularly for linguistically distant language pairs, and contribute to more accurate, fluent, and semantically rich machine-generated translations.

# BACKGROUND AND RELATED WORK

## Fundamental concepts of deep learning translation

In machine translation, a language model plays an important role in predicting the probability of a word or phrase to appear based on past context. Therefore, enhancing the quality of translation largely depends on developing a superior language model. NMT is an end-to-end learning system that translates an input sequence into an output sequence. The goal is to achieve accurate prediction of the target sequence from the input source as a high-level classification problem that places sentences in a shared semantic space. Given a parallel *corpus C* of source and target sentence pairs $(x, y)$, training is focused on maximization of the likelihood function $L$ over the model parameters $q$:

$$L_q = \sum_{(x,y)\in C} \log p(y|x; q), \tag{1}$$

where $x = x_1, \ldots, x_n$ is an input sentence, $y = y_1, \ldots, y_m$ is its translation, and $q$ is the learnable set of parameters. The conditional probability of the target sequence $y$ given $x$ is computed as:

$$p(y|x; q) = \prod_{j=1}^{m} p(y_j|y_{<j}, x; q), \tag{2}$$

where $y_j$ is the word being output currently, and $y_{<j}$ are the previously generated words. Beam search is typically employed during inference to get the most probable translation.

### Sequence-to-sequence framework

The most common method in NMT is the Sequence-to-Sequence (Seq2Seq) framework, as shown in Fig. 1. Here, the encoder processes an input sentence sequentially, mapping words to embeddings and fine-tuning them into contextualized representations. The encoded representations have the sentence's meaning and are passed to the decoder. The decoder used to generate words, constructs new representations that determine the next output word. The encoder and decoder architectures can be recurrent neural networks (RNNs) and convolutional neural networks (CNNs).

In the Seq2Seq framework, the encoder processes an input sequence $S = (s_1, s_2, \ldots, s_N)$ and encodes it into a context representation $z$. The decoder then generates the corresponding output sequence $T = (t_1, t_2, \ldots, t_M)$ using $z$ as a reference. The encoder computes a sequence of hidden states as follows:

$$h_n = \phi(W_e s_n + U_e h_{n-1} + b_e), \tag{3}$$

where $h_n$ represents the hidden state at step $n$, $W_e, U_e, b_e$ are trainable parameters, and $\phi$ is an activation. In the decoder block, each output word is generated iteratively based on previous outputs and the context:

$$d_m = \phi(W_d t_{m-1} + U_d d_{m-1} + b_d) \tag{4}$$

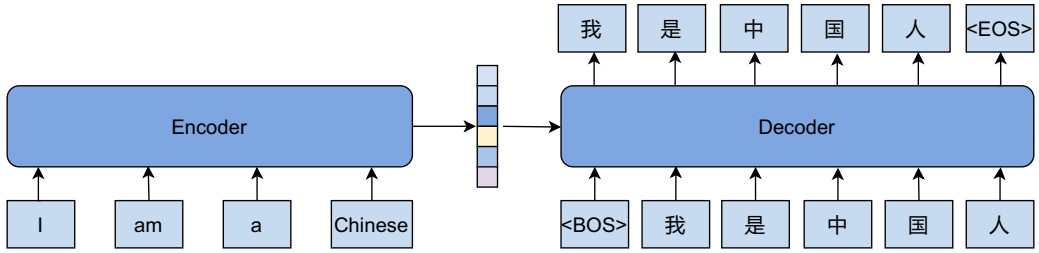

**Figure 1 Overview of the sequence-to-sequence framework for machine translation.**

where $d_m$ denotes the decoder's hidden state at step $m$. $W_d, U_d, b_d$ are trainable parameters, and $\phi$ is an activation.

### Attention mechanism

Recently, fully attention-driven NMT models have demonstrated exceptional performance, with attention mechanisms evolving from an auxiliary role to a primary means of extracting text features. A prime example of this evolution is the Transformer (*Vaswani et al., 2017*) architecture, which relies entirely on self-attention and feed-forward layers. Unlike RNNs or CNNs, the Transformer processes entire sentences simultaneously, rather than sequentially, resulting in more efficient and accurate translations. Stacking multiple Transformer layers further enhances translation quality. In self-attention layers, attention is computed across both the encoder and decoder. The alignment score, which determines how well different input positions align with a target position, is calculated as follows:

$$e_{ji} = a(s_{j-1}, h_i), \quad a_{ji} = \frac{\exp(e_{ji})}{\sum_{k=1}^{m} \exp(e_{ki})}, \tag{5}$$

where $e_{ji}$ is the alignment score and $a$ is a function that calculates how well an input word at position $i$ aligns with an output word at position $j$. These calculations are influenced by the decoder's previous step hidden state, $s_{j-1}$, and the encoder's hidden state, $h_i$. The attention weights are applied to the encoder's hidden states to generate a context vector:

$$c_j = \sum_{i=1}^{n} a_{ji} h_i. \tag{6}$$

This context vector, along with the previous hidden state and generated word, helps compute the next decoder hidden state:

$$s_j = g(s_{j-1}, y_{j-1}, c_j), \tag{7}$$

where $g$ is an activation function, $y_{j-1}$ is the embedding of the previous word, and $s_{j-1}$ is the last decoder state. Finally, the next target word is predicted using a feed-forward and softmax layer:

$$P(y_j \mid y_{<j}, x) = \text{softmax}(f(s_j, y_{j-1}, c_j)). \tag{8}$$

## Related work

The evolution of NMT has been marked by significant advancements in model architectures, training techniques, and the incorporation of linguistic features. Sequence-to-sequence architectures have been the dominant architecture for NMT models so far, mapping the source language into a fixed-dimensional representation and then transforming it into the target language. This foundational approach has been enhanced by attention mechanisms to allow models to focus on relevant parts of the input sequence during translating and thus improving the quality of the translation (*Klein et al., 2017*; *Cheng et al., 2018*). One of the primary challenges of English-Chinese and Chinese-English translation is due to the structural differences between the two languages. English is an alphabetic language with a subject-verb-object (SVO) structure, while Chinese is logographic and has a tendency to employ a SVO or topic-comment structure. This tends to result in word alignment and sentence reordering issues in translation. *Zhang et al. (2017)* has shown that the incorporation of linguistic features such as syntactic forms and word reordering ability has the ability to enhance translation to a great extent. For instance, *Sennrich & Haddow (2016)* demonstrated that augmenting NMT systems with linguistic input features improved their ability to handle complex sentence structures, thereby reducing errors in translation.

Access to large-scale parallel corpora has also played a crucial role in pushing English-Chinese NMT systems. The WCC-EC 2.0 *corpus* contains over 1.6 million English-Chinese sentence pairs and it is one such contribution of large-scale data towards translation performance. This web-crawled *corpus* has been shown to improve the robustness and accuracy of NMT models, with high BLEU scores achieved in test datasets (*Zhang et al., 2024*). Moreover, *Wang et al. (2021)* showed that the use of monolingual data in addition to parallel corpora has been explored as a way to further enhance translation quality, particularly in low-resource settings. In 2023, *Wang et al. (2023a)* engaged in attempting to address some of the linguistic phenomena that complicate translations, such as the pro-drop feature of Chinese. Wang's study on a pronoun omission solution recognizes the challenge posed by languages that have a tendency to drop pronouns, causing fragmented translations when they are translated into English. This work highlights the necessity of developing tailored models that have the ability to complete missing information in order to produce fluent translations.

Recently, the developments in model architecture such as the Transformer model have significantly improved the performance and efficiency of NMT systems. *Meng & Zhang (2019)* applied Transformer architecture, founded on self-attention mechanisms to allow handling of long-distance word relations within sentences, and hence it is best suited to the intricacies of English-Chinese translation. *Hassan et al. (2018)* demonstrated that Transformer-based models outperform traditional RNNs on a variety of translation tasks, including between English and Chinese. The use of character-level representations has also been proven to be an efficient approach to enhancing NMT systems for Chinese. *Nikolov*

*et al. (2018)* and *Zhang & Komachi (2018)* illustrated that character-level models have the ability to capture the nature of Chinese characters, leading to more precise English-Chinese translations. This approach addresses the challenge posed by the lack of systematic correspondence between the linguistic units of two languages. Moreover, research into multilingual NMT systems has been increasing with the capacity for knowledge transfer across language pairs. This is particularly beneficial for low-resource languages where parallel data could be scarce. By leveraging data from different languages, *Dabre, Chu & Kunchukuttan (2020)* have shown that NMT systems are able to perform better in English-to-Chinese and other language pair translations. In terms of evaluation metrics, the quest for human parity in translation quality has motivated significant research efforts. Studies have been made to define and measure human parity in translation, with some NMT systems already found to reach the levels of professional human translators in some settings, such as Chinese-English news translation by *Hassan et al. (2018)*. Such accomplishment highlights the potential of NMT to enhance cross-cultural communication and interaction and develop cross-cultural interaction.

# MATERIAL AND METHOD

## BMELD dataset

In this study, we focus on the Bilingual MELD (BMELD) (*Liang et al., 2021*) dataset, a novel resource designed to enhance the translation quality between English and Chinese. Building upon the existing MELD *corpus*, which comprises monolingual English dialogues, BMELD introduces a multimodal dialogue dataset that integrates emotional and sentiment annotations. Each entry in the dataset includes detailed information, such as speaker identity, emotional state, sentiment classification, and contextual details from the popular TV show "Friends". The dataset aims to balance the representation of speakers, with 50% of the dialogues provided in Chinese, ensuring a diverse linguistic context. The Chinese translations were meticulously crawled and manually post-edited by native speakers, guaranteeing accuracy and fluency. By leveraging the Stanford CoreNLP toolkit for sentence segmentation, BMELD facilitates a robust framework for analyzing and improving translation methodologies. Our research utilizes this rich dataset to explore advancements in translation techniques, particularly in handling emotional nuances and context-specific dialogue, ultimately contributing to more sophisticated bilingual translation systems.

### Data preprocessing

The preprocessing of bilingual data follows a structured pipeline approach to ensure high-quality input for the translation model. This process involves multiple steps, including data loading, tokenization, vocabulary generation, sequence encoding, and batch preparation. The dataset consists of parallel English-Chinese sentence pairs, with each line containing an English sentence alongside its corresponding Chinese translation. To facilitate model training, tokenization is applied to each sentence, incorporating special tokens: BOS (Beginning of Sentence) and EOS (End of Sentence). English sentences

undergo lowercasing before tokenization, while Chinese sentences are segmented at the character level.

To enhance neural network training, separate vocabularies are constructed for English and Chinese. The most frequent words, up to a maximum of 30,000, are retained, whereas less frequent words are substituted with a special UNK (unknown) token. Each token is then mapped to a unique numerical identifier, forming a word-to-index dictionary. A reverse index is also established to facilitate decoding. Sentences are subsequently converted into sequences of numerical IDs according to the constructed dictionaries. To improve training efficiency, sentences are sorted by length before batching, thereby minimizing unnecessary padding. For mini-batch training, sentences are grouped into fixed-size batches. As sentence lengths may vary within a batch, padding is applied to standardize their length. Additionally, a mask is generated to differentiate actual tokens from padding elements. This step ensures that the model processes data efficiently while reducing redundant computations on padded tokens. Finally, the dataset is split into 10,477 sentences for training, 1,177 for validation, and 2,763 for testing. This stratification ensures that the model has sufficient data to learn while allowing for performance tuning and unbiased evaluation on unseen examples.

## Model development

### Kolmogorov-Arnold networks

The Kolmogorov-Arnold network (KAN) is a neural network architecture inspired by the Kolmogorov-Arnold representation theorem. This theorem states that any multivariate continuous function defined on a bounded domain can be expressed as a finite composition of continuous univariate functions and addition. This leads to a novel approach in neural network design, where nonlinear transformations are performed through learnable univariate functions. In KAN, each layer consists of a set of learnable univariate functions acting as activation functions on the edges. Specifically, a Kolmogorov-Arnold layer with $d_{\text{in}}$-dimensional inputs and $d_{\text{out}}$-dimensional outputs is defined as:

$$f(\mathbf{x}) = \Phi \circ \mathbf{x} = \left[ \sum_{i=1}^{d_{\text{in}}} \phi_{i,1}(x_i) \quad \cdots \quad \sum_{i=1}^{d_{\text{in}}} \phi_{i,d_{\text{out}}}(x_i), \right] \tag{9}$$

where $\Phi$ is a matrix of univariate functions $\phi_{i,j}$, to enable a learnable nonlinear mapping of the input. A KAN is built by stacking several such layers, resulting in sequential compounding of transformations:

$$KAN(\mathbf{x}_0) = \Phi_{L-1} \circ \Phi_{L-2} \circ \cdots \circ \Phi_0 \circ \mathbf{x}_0. \tag{10}$$

To enhance the flexibility of the learnable univariate functions, KAN parameterizes $\phi(x)$ as a combination of the SiLU activation function and a B-spline function:

$$\phi(x) = w_b \cdot \text{silu}(x) + w_s \cdot \text{spline}(x), \tag{11}$$

where

$$\text{silu}(x) = \frac{x}{1 + e^{-x}}, \quad \text{spline}(x) = \sum_i c_i B_i(x). \tag{12}$$

### Justification for model choice

Typical NMT models, such as Transformer-based architectures, have achieved remarkable success in English-Chinese translation. However, they perform poorly in capturing long-range dependencies, handling complex syntactic structures, and maintaining translation consistency across linguistically distant language pairs. Studies based on the self-attention mechanism allow models to capture global context; however, they do not effectively model hierarchical or compositional linguistic structures, which often results in inconsistent translations. In addition, most previous models still face challenges in computational efficiency as well as accuracy when translating long documents or operating under high-speed processing demands. To address these limitations, we employ the Kolmogorov-Arnold Transformer (KAT), a novel architecture whose purpose is to boost the expressiveness and efficiency of deep learning models in sequence-to-sequence tasks.

KAT is a hybrid architecture that integrates the strengths of Transformers and Kolmogorov–Arnold Networks, enabling the representation of complex functions through a structured decomposition process. This capability is particularly advantageous in machine translation, where capturing intricate relationships between source and target languages is essential. As a result, KAT can handle long sentences in both English and Chinese, enhancing the model's ability to produce accurate and contextually coherent translations. This architecture improves parameter efficiency and reduces the risk of overfitting while maintaining high fidelity in translation. It is especially adept at capturing syntactic and semantic relationships in Chinese due to challenges such as word segmentation and character-based representation. By leveraging KAT, we aim to enhance translation quality through improved contextual understanding, reduced computational complexity, and greater fluency in both English and Chinese. This choice offers a robust, scalable, and interpretable approach to machine translation, advancing the capabilities of current NMT systems.

### Model architecture

In this article, we propose the customed transformer architecture by replacing FFN with a KAN layer to enhance the expressiveness of the transformation. Overview of this architecture is described in Fig. 2.

Given an input sequence $X$ consisting of word-tokenized representations, discrete tokens are initially mapped into dense vector spaces through an embedding layer. Positional encodings are incorporated to preserve order information, as the Transformer model does not inherently capture positional dependencies:

$$X = \text{EmbeddingLookup}(X) + \text{PositionalEncoding}(X), \tag{13}$$

where $X \in \mathbb{R}^{\text{batch\_size} \times \text{seq\_len} \times \text{embed\_dim}}$. The core component of the Transformer is the self-attention mechanism, which enables the model to capture long-range dependencies

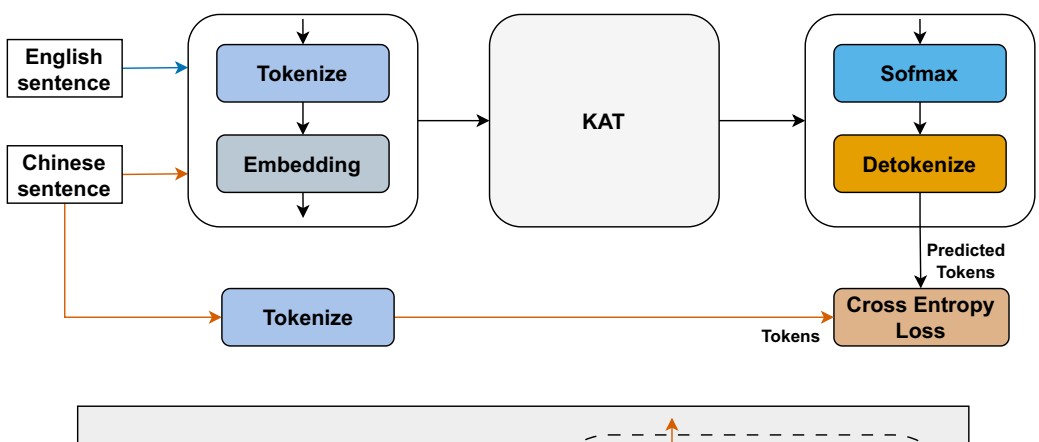

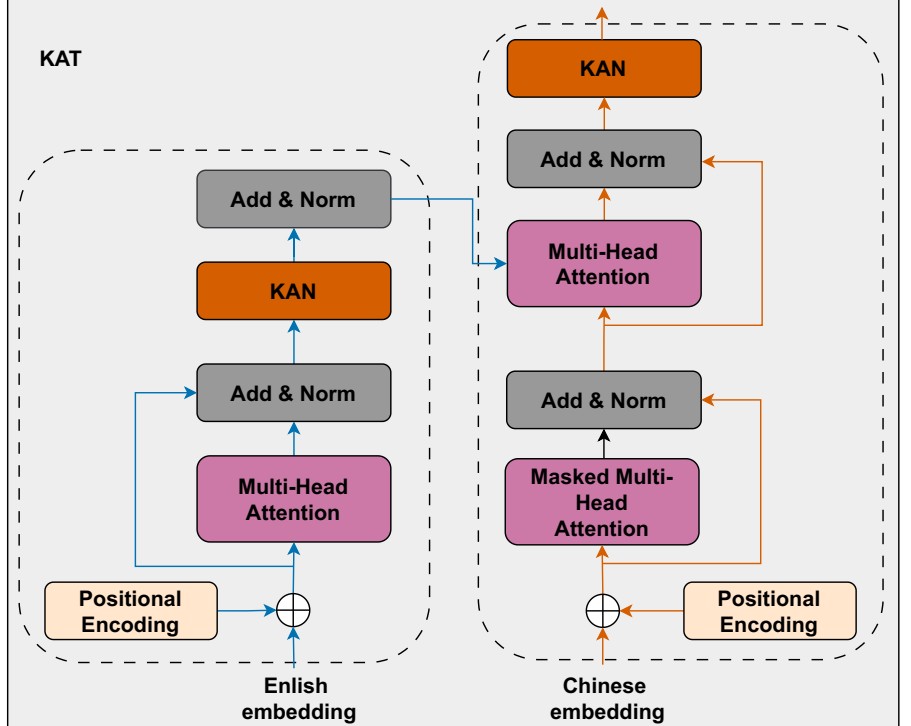

**Figure 2 KAT model: architectural overview and implementation details.**

between words. The input $X$ is projected into Query ($Q$), Key ($K$), and Value ($V$) representations. These representations are then used to compute the self-attention output:

$$\alpha = \text{SelfAttention}(Q, K, V), \quad Q = XW_Q, \quad K = XW_K, \quad V = XW_V, \tag{14}$$

where $W_Q, W_K, W_V$ are learnable weights. To facilitate stable training and prevent gradient vanishing, we incorporate residual connections followed by layer normalization:

$$X_{\text{norm}} = \text{LayerNorm}(X_{\text{attention}} + \alpha). \tag{15}$$

Next, each token representation is then passed through a KAN that is explained in next subsection and a final residual connection and normalization step is applied:

**Table 1 The adjustable training settings.**

| Parameters | Description | Default value |
|---|---|---|
| src_vocab_size | English vocabulary size | 30,522 |
| tgt_vocab_size | Chinese vocabulary size | 21,128 |
| batch | Batch size | 128 |
| num_epochs | Number of epochs | 30 |
| hidden_dimensions | Feature dimensions of the model | 256 |
| n_head | Number of attention head of model | 4 |
| num_encoder_layers | Number of encoder layers | 2 |
| num_decoder_layers | Number of decoder layers | 1 |
| optimizer | The optimizer for training | Adam |

$$X_{\text{hidden}} = \text{KAN}(X_{\text{norm}}). \tag{16}$$

At the end of multiple encoder layers, the output representation $X_{\text{out}}$ captures rich contextual information, which is then passed to the decoder for translation.

$$X_{\text{out}} = \text{LayerNorm}(X_{\text{hidden}} + X_{\text{norm}}). \tag{17}$$

This structured processing allows the model to effectively encode syntactic and semantic dependencies, making it well-suited for NMT tasks.

## Implementation details

Our model is implemented using PyTorch. The network undergoes training with a batch size of 128 and utilizes a cross-entropy loss function over 50 epochs. The training process is conducted on a GPU 3070 with 24 GB of RAM. We employ the Adam optimizer with a learning rate of 0.0001, momentum parameters $b_1 = 0.9$ and $b_2 = 0.98$, and $\varepsilon = 10^{-9}$. The chosen hyperparameters are summarized in Table 1.

Our model is implemented using PyTorch. The network undergoes training with a batch size of 128 and utilizes a cross-entropy loss function over 50 epochs. The training process is conducted on a GPU 3070 with 24 GB of RAM and an Intel Core i9-12900K 16-core CPU, 32 GB of system RAM, and running Ubuntu 22.04 LTS. The hyperparameters during training were chosen based on a balance between learning efficiency and the system's practical computational capacity. A batch size of 128 maximizes the GPU's parallel processing capability while ensuring stable updates of the network weights. Training over 50 epochs was determined through prior experimentation to give the model sufficient time to converge without overfitting. The learning rate of 0.0001 was selected after multiple trials with different learning rates, optimizing for the best performance. Along with the momentum parameters $b_1 = 0.9$, $b_2 = 0.98$, and $\varepsilon = 10^{-9}$, these settings were chosen for their ability to automatically adjust the learning step, helping the model converge quickly and stably on the dataset. These parameters are also recommended in many recent studies on similar deep learning models, as they help prevent gradients from becoming too large or too small. The chosen hyperparameters are summarized in Table 1. On this setup, the average training time per epoch is approximately 2.4 min, resulting in a

total training time of around 2 h for the full 50 epochs. During testing, the average inference time per sample is approximately 2.1 s.

## Evaluation method

### Evaluation metrics

In this study, we use evaluation metrics including Bilingual Evaluation Understudy (BLEU), METEOR, Translation Edit Rate (TER), and Character-level F-score (ChrF) to assess the performance of our translation models. BLEU focuses on n-gram overlap between the translation and reference, making it suitable for overall evaluation but potentially overlooking semantic and contextual aspects. METEOR supplements this by incorporating synonym matching and sentence-level analysis, providing a better reflection of translation accuracy and fluency. TER measures the number of edits required to transform the system output into the reference translation, offering insight into post-editing effort and specific differences. ChrF, based on character-level F-score, enables finer-grained evaluation of minor errors and is especially suitable for languages with complex word structures. Comparing and combining these metrics offers a more comprehensive perspective on translation quality, thereby enhancing the accuracy of model performance assessment. They characterize their mathematical formulas as:

$$\text{BLEU} = \exp\left(\sum_{n=1}^{N} w_n \log p_n\right) \times \min(1, e^{1-r/c}), \tag{18}$$

where $p_n$ represents the precision of n-grams, $w_n$ is the weight assigned to each n-gram, $r$ is the reference translation length, and $c$ is the candidate translation length.

$$\text{METEOR} = F_{\text{mean}} \times (1 - \text{penalty}), \tag{19}$$

where $F_{\text{mean}}$ is a weighted harmonic mean of precision $P$ and recall $R$, and the penalty term accounts for the number of chunks that are not in the correct order.

$$\text{TER} = \frac{\text{E}}{\text{R}}, \tag{20}$$

where $E$ represents the number of insertions, deletions, substitutions, and shifts needed, and $R$ is the total number of words in the reference translation. A lower TER score indicates a more accurate translation.

$$\text{ChrF} = (1 + \beta^2) \cdot \frac{\text{Precision} \times \text{Recall}}{\beta^2 \times \text{Precision} + \text{Recall}}, \tag{21}$$

where Precision and Recall are calculated over character-level n-grams, and $\beta$ is a parameter balancing the two measures.

### Comparative analysis

Following the guidelines for comparative studies on various network models, the hardware and software environments, along with other experimental parameters, remain constant, with the network model being the only variable. This study employs a statistical machine translation model (*Osborne, 2011*) as the baseline system. In addition, various neural

network architectures, including RNNs (*Datta et al., 2020*), long short-term memory (LSTM) networks (*Ramaiah, Datta & Agarwal, 2022*), and bidirectional LSTM (BiLSTM) (*Tanvir et al., 2023*) models, hybrid models such as RNN+Attention (*Shi, Meng & Wang, 2019*), LSTM+Attention (*Wu & Xing, 2024*), convolutional sequence to sequence learning (CSSL) (*Gehring et al., 2017*), BART (*Lewis et al., 2019*) and Transformer (*Badawi, 2023*) model are also analyzed and evaluated. These models developed under the same environment and parameter settings.

The training configurations for all baseline models were kept consistent to ensure fair comparison. Specifically, we used the Adam optimizer with a learning rate of 1e−4, a batch size of 128. The hidden dimension was set to 256, and for attention-based models (Transformer, BART), we used four attention heads. Encoder-decoder models (RNN, LSTM, CSSL) followed a 2-layer encoder and 1-layer decoder architecture. These settings were applied uniformly unless otherwise specified for a particular model.

### Benchmarking KAN integration
To assess the impact of incorporating KAN into the KAT architecture, we conducted comparative experiments against a baseline Transformer model. The models were evaluated on the Bilingual MELD dataset using standard translation quality metrics, including BLEU-4, METEOR, TER, and ChrF. BLEU-4 measures n-gram precision against reference translations, METEOR evaluates alignment based on synonymy and word order, TER quantifies the number of edits needed to match the reference, and ChrF assesses character-level translation quality. Performance improvements across all four metrics were used to validate the effectiveness of the KAN integration.

## Evaluating the impact of vocabulary size
To investigate the effect of vocabulary size on translation performance, we trained KAT models with varying vocabulary sizes of 10, 20, and 30 K. The models were evaluated on the Bilingual MELD dataset using BLEU-4, METEOR, TER, and ChrF metrics. BLEU-4 and METEOR scores were used to assess translation fluency and adequacy, while TER measured the amount of post-editing required, and ChrF evaluated character-level accuracy. Performance trends across different vocabulary sizes were analysed to determine the trade-offs between translation quality and computational efficiency.

## RESULTS AND DISCUSSION
### Comparative results
The results in Table 2 demonstrate that neural network-based models significantly outperform the SMT baseline across all evaluation metrics. The BLEU-4 score for the baseline system is 21.5, which is considerably lower than that of the RNN-based models and even more so compared to transformer-based architectures. Among the recurrent neural network models, BiLSTM exhibits the best performance with a BLEU-4 score of 30.5. When the attention mechanism is incorporated, both RNN+Attention and LSTM+Attention show notable improvements, with the latter achieving 34.8 in BLEU-4 and 38.2 in METEOR, highlighting the effectiveness of attention mechanisms in translation tasks.

**Table 2 Performance comparison of different translation models on the test set.**

| Model | BLEU-4 | METEOR | TER | ChrF |
|---|---|---|---|---|
| Statistical MT (Baseline) | 21.5 | 25.3 | 60.4 | 45.7 |
| RNN | 26.7 | 30.1 | 55.2 | 50.5 |
| RNN + Attention | 32.1 | 35.5 | 49.8 | 56.0 |
| LSTM | 28.9 | 32.4 | 53.1 | 52.8 |
| BiLSTM | 30.5 | 34.2 | 51.3 | 54.3 |
| LSTM + Attention | 34.8 | 38.2 | 47.5 | 58.3 |
| Transformer | 39.2 | 41.7 | 43.8 | 62.1 |
| CSSL | 37.1 | 39.8 | 45.2 | 60.3 |
| BART | 40.6 | 42.3 | 42.6 | 62.9 |
| Kolmogorov-Arnold Transformer | 42.8 | 45.3 | 40.5 | 65.4 |

The standard Transformer model further enhances translation quality, surpassing RNN-based models by a large margin with a BLEU-4 score of 39.2. This improvement aligns with previous research, confirming the efficiency of self-attention mechanisms in capturing long-range dependencies in sequential data. Notably, the Kolmogorov-Arnold Transformer achieves the highest performance across all metrics, with a BLEU-4 score of 42.8 and a significant reduction in TER (40.5). These results suggest that incorporating Kolmogorov-Arnold transformations allows for more effective feature extraction and representation learning in translation tasks. The improvement in ChrF (65.4) further supports that this model in handling character-level details, which is crucial for morphologically rich languages such as Chinese. In addition, the CSSL-based model achieves competitive results, with a BLEU-4 score of 37.0 and a METEOR score of 39.8, compared to BART's scores of 40.6 (BLEU-4), 42.3 (METEOR). To directly address the effectiveness of KAT in long-sequence translation, we provide several representative examples from the test set in Table 3. These qualitative results highlight specific improvements made by KAT compared to conventional models, particularly in preserving long-range dependencies and maintaining translation coherence across extended sequences. This analysis further supports the claim that KAT is a promising approach for bilingual machine translation tasks involving long-form content.

## Impact of KAN

Table 4 presents the performance metrics for different configurations of KAN-enhanced translation models. The results demonstrate that KAN improves translation fluency and adequacy by leveraging a more sophisticated representation of the input data.

The incorporation of KAN leads to a BLEU-4 score increase from 39.2 to 44.3, which highlights its ability to better capture linguistic details. Furthermore, the METEOR score improves significantly, indicating better alignment with human translation references. The reduction in TER suggests fewer necessary post-editing changes, and the increase in ChrF indicates superior handling of character-level variations, which is particularly beneficial for complex language pairs. Theoretically, KAT builds upon the foundation of KAN, which employs Kolmogorov-Arnold representations to enhance the approximation of complex

**Table 3 Example translations generated by different models for qualitative comparison.**

| Source sentence | Target sentence | Transformer | CSSL | BART | Ours |
|---|---|---|---|---|---|
| Long sentences | | | | | |
| Why do all your coffee mugs have numbers on the bottom? | 咖啡杯底为什么都有编号? | 为什么你的咖啡杯底下都有数字? | 你的咖啡杯底下怎么都有编号? | 你的咖啡杯为什么底下有数字? | 你的咖啡杯底为什么都有编号? |
| Oh. That's so Monica can keep track. That way if one on them is missing, she can be like, Where's number 27?! | 这样莫妮卡才方便追踪. 要是少了一个, 她就可以说"27号杯呢? | 哦, 这样莫妮卡可以知道哪个不见了, 她就会问: 27号在哪? | 这样莫妮卡能追踪, 如果少了一个, 她就会说, 27号呢? | 哦, 这样莫妮卡就能知道少了哪个, 她会说: 27号去哪了? | 哦, 这样莫妮卡可以追踪. 这样如果少了一个, 她可以说, 27号呢? |
| Short sentences | | | | | |
| You know what? | 你知道吗? | 你知道吗? | 你知道吗? | 你知道吗? | 你知道吗? |
| Come on, Lydia, you can do it. | 加油, 莉迪亚, 你办得到. | 加油, 莉迪亚, 你一定可以. | 加油, 莉迪亚, 你能做到. | 加油, 莉迪亚, 你行的. | 加油, 莉迪亚, 你可以的. |
| Push them out, push them out, harder, harder. | 用力, 再加油. | 把他们推出去, 再用点力. | 推出去, 用力点, 更用力. | 用力, 推出去, 更用力! | 把他们推出去, 再用力! |
| Let's get, I was just. Yeah, okay. | 我们, 我只是……好吧. | 我们, 我只是…好吧. | 我们, 我只是…好吧. | 我们, 我只是…好吧. | 我们, 我只是…好吧. |

**Table 4 Effect of KAN on translation performance.**

| Model | BLEU-4 | METEOR | TER | ChrF |
|---|---|---|---|---|
| Transformer | 39.2 | 41.7 | 43.8 | 62.1 |
| Kolmogorov-Arnold transformer | 42.8 | 45.3 | 40.5 | 65.4 |

nonlinear functions. This approach enables the model to learn deeper and more structured semantic representations from the input data. Specifically, KAT improves the Transformer's ability to capture nonlinear dependencies and compositional structures—fact attention mechanisms to model effectively. As a result, it produces more accurate contextual representations, thereby improving the fluency and semantic adequacy of translations, as reflected in the BLEU, METEOR, TER, and ChrF scores. These findings reinforce the potential of KAN as an effective enhancement to existing transformer-based translation models, offering a promising direction for future research in neural machine translation.

### Impact of vocabulary size

Table 5 shows the impact of different vocabulary sizes on model performance. The results indicate that increasing vocabulary size leads to improvements in translation quality, as evidenced by higher BLEU-4 and METEOR scores and reduced TER. However, beyond 30 K vocabulary size, the improvements become marginal while computational costs increase significantly. Thus, selecting an optimal vocabulary size is essential to balancing performance and efficiency in machine translation models.

**Table 5 Effect of vocabulary size on translation performance.**

| Vocabulary size | BLEU-4 | METEOR | TER | ChrF |
|---|---|---|---|---|
| 10 K | 35.1 | 38.9 | 48.2 | 57.1 |
| 20 K | 38.7 | 41.2 | 44.5 | 60.3 |
| 30 K | 43.5 | 46.0 | 40.2 | 64.8 |

## LIMITATIONS AND FUTURE RESEARCH

Despite the promising performance of our approach, several limitations warrant attention. First, while the Bilingual MELD dataset serves as a useful benchmark, it may not sufficiently capture the full diversity of linguistic structures, idioms, and domain-specific expressions encountered in real-world translation tasks. This could hinder the generalizability of KAT across broader language settings. Second, the high computational demands of KAT pose scalability challenges. Training and inference require substantial hardware resources, which could limit accessibility for low-resource environments. Another limitation involves the reliance on automated metrics (BLEU, METEOR, TER), which do not fully capture translation fluency or semantic adequacy. Future studies should incorporate human evaluation to better understand fluency, adequacy, and contextual appropriateness.

Future extensions of KAT will explore multilingual generalisation by integrating insights from domain adaptation, such as multilevel distribution alignment strategies proposed by *Ning et al. (2025)*, and dynamic label alignment techniques exemplified in DyLas by *Ren et al. (2025)*. These approaches may enhance KAT's robustness across diverse linguistic domains and shifting data distributions. Incorporating continual learning mechanisms, such as the dual-channel collaborative transformer developed by *Cai et al. (2025)*, could help prevent catastrophic forgetting when fine-tuning on evolving bilingual corpora. Additionally, contrastive representation learning, as demonstrated by *Nguyen et al. (2024)* in multimodal molecular property prediction, offers a compelling strategy for refining latent language representations. Expanding KAT to support multimodal translation—especially in video and dialogue contexts—may benefit from attention-based architectures like TASTA (*Wang et al., 2023b*) and generation models like AttriDiffuser (*Song et al., 2025*), which capture spatial-temporal and semantic alignment in text-to-image tasks. Such techniques could improve context awareness in complex, real-world translation scenarios. Moreover, incorporating domain-specific linguistic features, inspired by findings in Chinese EFL learner studies (*Zhang & Chen, 2024*), could refine KAT's handling of culturally and structurally distinct expressions. Large language model augmentation, as explored in the Fg-T2M++ framework by *Wang et al. (2025)*, may further support nuanced text generation in translation. Lastly, the application of learned molecular representations in bio-cheminformatics (*Nguyen-Vo et al., 2024*) suggests promising parallels in encoding structured language information, which KAT could adopt to enrich its vocabulary modelling and translation consistency over longer sequences.

## CONCLUSIONS

This article presented a novel architecture that integrates the Kolmogorov-Arnold network into a Transformer framework to improve English-Chinese machine translation. Evaluated on the Bilingual MELD dataset, the proposed model achieved superior performance compared to both traditional models and modern neural translation systems, with a BLEU-4 score of 42.8, a METEOR score of 45.3, and a TER of 40.5. These results highlight the effectiveness of incorporating structured mathematical representations into neural architectures. Furthermore, we showed that increasing vocabulary size and incorporating Kolmogorov-Arnold-based components contributed to notable gains in translation accuracy. Overall, this study demonstrates the potential of the framework as a powerful approach for enhancing the quality of bilingual translation systems.

### Funding

The author received no funding for this work.

### Competing Interests

The author declares that they have no competing interests.

### Author Contributions

- Yuzhe Nie conceived and designed the experiments, performed the experiments, analyzed the data, performed the computation work, prepared figures and/or tables, authored or reviewed drafts of the article, and approved the final draft.

### Data Availability

Data is available at GitHub: https://github.com/XL2248/CPCC.

Code is available the Supplemental Files.

### Supplemental Information

Supplemental information for this article can be found online at http://dx.doi.org/10.7717/peerj-cs.3139#supplemental-information.

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
