# Peer review of "Improving English-Chinese translation using the Kolmogorov-Arnold Transformer"

_PeerJ Computer Science, doi:10.7717/peerj-cs.3139_

## Round 0.1 · original submission · Major Revisions

Please consider the comments from the reviewers and revise the manuscript accordingly.

Reviewer 1 ·

Basic reporting

This paper addresses a challenge in machine translation by proposing the Kolmogorov-Arnold Transformer (KAT). The introduction effectively contextualizes the need for improved translation quality and highlights the limitations of existing models. The background section appropriately cites relevant literature. Experimental results are well-documented, showing clear improvements in BLEU, METEOR, and TER metrics. All technical terms are clearly defined, and the overall methodology is coherent. The code and data are fully provided. Key components such as evaluation metrics and data preprocessing are well described. I have only one minor comment on the paper, as outlined below.

Experimental design

The authors should compare their method with CSSL (https://arxiv.org/pdf/1705.03122v3) as it is a good and suitable method for this topic.

Validity of the findings

Some more details could improve the quality of the manuscript, for example:
- How long did the model take to train for 1 epoch? Including wall-clock time or number of epochs would provide a useful sense of computational cost.
- Bief description of the hardware used, such as the CPU model, amount of RAM, and operating system?
- Provide information such as the number of encoder/decoder layers, hidden dimensions, attention heads, etc… to support reproducibility and insight.

Cite this review as

Reviewer 2 ·

Basic reporting

- The introduction clearly explains the motivation for improving cross-lingual understanding and positions KAT as an advancement over existing models.
- The background section is comprehensive, with sufficient references and high-quality figures and tables.
- The manuscript meets academic standards in clarity, formality, and presentation.

Experimental design

- More detail is needed on the dataset split and the training configurations for baseline models (e.g., optimizer, learning rate, batch size).
- Please include explicit formulas for the SiLU activation function and the Spline component to aid reader's understanding.
- Reporting the average inference time during testing would help assess real-world applicability.

Validity of the findings

- Consider including a comparison with BART, as it is a strong baseline for various NLP tasks.

Cite this review as

Reviewer 3 ·

Basic reporting

* The manuscript is well-written and generally adheres to professional language standards. However, some sentences are overly complex and could benefit from simplification for clarity.
* The purposes of this paper are clearly defined in the introduction. The authors emphasize the importance of high-quality English-Chinese translation in global communication and mention the challenges in this matter.
* The author has cited relevant literature. The author also presents the state of the art of recent studies and focuses on the method based on the Traditional SMT and NMT systems. However, the author may elaborate on the drawbacks of the recent study to emphasize the novel solution of the paper.
* The paper structure already follows the PeerJ standards, though clearer section transitions would improve readability.

Experimental design

* The experiment conducted in the paper falls within the journal's scope and presents the solution of a novel framework using the Kolmogorov-Arnold Transformer (KAT). The experiment is conducted using the MELD dataset.
* The methods are generally rigorous, but justification for hyperparameter selection (such as learning rate and batch size) would add credibility.
* Evaluation metrics (BLEU, METEOR, TER, ChrF) are appropriate, but additional discussion comparing their significance in assessing translation quality would enhance the study.

Validity of the findings

* The results indicate that KAT outperforms traditional models, providing evidence of improved translation accuracy.
* Comparisons to other models are informative, yet further justification is needed for why KAT surpasses standard Transformers beyond quantitative results.
* Some claims regarding KAT’s effectiveness in long-sequence translation require more direct validation, particularly through qualitative analysis or examples showcasing specific improvements.
* The conclusion summarizes findings effectively but could better highlight the limitations and potential future directions of the model.

Additional comments

* The manuscript presents a compelling contribution to machine translation research, leveraging an innovative approach.
* Additional discussion on real-world applications of KAT outside research contexts (e.g., commercial use, integration into existing translation software) would broaden its impact.
* The figures and tables are generally clear, but adding annotations or descriptions could further enhance comprehension.
* While the study demonstrates improvements in translation quality, a human evaluation component (such as expert linguist assessments) would strengthen its claims.

Cite this review as

---

## Round 0.2 · Minor Revisions

Although the authors have addressed all reviewer comments, the Introduction still lacks a clear statement of the specific problem this study tackles and does not adequately highlight the main contribution. For improved clarity and coherence, I recommend concluding the Introduction with a concise paragraph that summarizes the novelty of the proposed approach and its intended impact.

Reviewer 1 ·

Basic reporting

The revised version is well-organized and clearly written. I don't have any other comments.

Experimental design

I have no further comments.

Validity of the findings

I have no further comments.

Additional comments

No further comments

Cite this review as

Reviewer 2 ·

Basic reporting

- The revised manuscript is clear, well-organized, and aligns with the journal's scope.
- The authors have provided detailed information on dataset splits, model configurations, hardware specifications, and training procedures, enhancing reproducibility and transparency.

Experimental design

- The authors have provided thorough details on dataset splits, model configurations, training procedures, and hardware specifications, with explicit architectural information.
- The manuscript now reports average training and inference times, offering valuable insight into computational cost and real-world applicability.

Validity of the findings

- As previously suggested, the authors have strengthened their results with comparisons to strong baselines, notably BART and CSSL.
- Evaluation metrics are sufficient, and including additional baselines improves the reliability of the reported improvements. Additional discussion on applications and future human evaluation further supports the study’s validity and impact.

Cite this review as

Reviewer 3 ·

Basic reporting

Thank you to the author who has already revised the paper according to my review. I find the introduction and background already clear, and the problem statements are easy to understand. The overall structure of the article conforms to standard academic paper norms, with clear sections for Introduction, Background and Related Work, Materials and Methods, Results and Discussion, Limitations and Future Research, and Conclusions. This logical flow enhances clarity and readability.

Experimental design

The revised version more clearly explains the experimental design. The investigation appears to have been conducted rigorously and follows the guidance of the journal. The methods used in the paper are easy to understand and technically sound. The evaluation method has shown the advantages of the proposed method.

Validity of the findings

The experiments and evaluations appear to be performed satisfactorily. The comparative results in Table 2 demonstrate the superior performance of the proposed Kolmogorov-Arnold Transformer (KAT) across all metrics (BLEU-4, METEOR, TER, ChrF) compared to various baseline models, including the standard Transformer. This supports the claim that incorporating Kolmogorov-Arnold transformations enhances feature extraction and representation learning. The conclusion answers all the questions mentioned in the introduction.

Additional comments

The paper presents a significant contribution to the field of English-Chinese machine translation by introducing the Kolmogorov-Arnold Transformer. The integration of KANs into the Transformer architecture is a novel approach that shows promising results. The experimental setup is robust, and the analysis of results is thorough.

Cite this review as

---

## Round 0.3 · accepted · Accept

The authors have addressed all of the comments. The manuscript is now ready for publication.